

# *Ex-situ* avian sex skews: determinants and implications for conservation

Clancy A. Hall[1], Gabriel C. Conroy[1,2] and Dominique A. Potvin[1,2]

[1] School of Science and Engineering, University of the Sunshine Coast, Petrie, QLD, Australia
[2] Centre for BioInnovation, University of the Sunshine Coast, Sippy Downs, QLD, Australia

## ABSTRACT

With over half of all avian species in decline globally, zoo-based recovery programs are increasingly relied upon to save species from extinction. The success of such programs not only rests with political will, but also in our understanding of species' breeding biology and how individuals and populations respond to changes in their environment. Sex skews, that is, an imbalance in the optimal number of males to females, is an underlying mechanism of population decline in some threatened species. *Ex-situ* (*i.e.*, zoo-based) management practices will need to become more efficient to support the growing number of conservation reliant species and manage sex skews to amend, repair and restore population stability both *in-* and *ex-situ*. In this article, we analysed data from over 182,000 birds in global *ex-situ* collections. We interpreted sex ratio variation by observing the proportion of males within and between orders, International Union for Conservation of Nature (IUCN) threat status and housing inside and outside of a species' natural range. Overall, our results showed that male-biased sex skews are more prevalent *ex-situ* than they are in the wild and although they vary greatly at the institutional level, were closer to parity at a global level. The variation amongst threat status and housing outside of range were less significant. These findings have implications for the conservation management of threatened birds and the potential of *ex-situ* populations to function with maximum effect in an integrated management system.

## INTRODUCTION

Sex ratios, that is, the ratio of males to females in a population, are considered a critical parameter influencing population growth and can undermine demographic potential when a skew to a particular sex goes beyond an optimal threshold (*Lambertucci et al., 2012*; *Székely et al., 2014*). Sex skews occur across a broad range of *in-* and *ex-situ* avian taxa and age cohorts, and are particularly prevalent in small and threatened populations (*Donald, 2007*; *Morrison et al., 2016*). It has recently been observed that sex ratios in wild plant and animal populations are changing as a response to anthropogenic effects such as global climate change, habitat loss, urbanisation and invasive species (*Capdevila et al., 2020*; *Katzner et al., 2020*; *Schacht et al., 2022*). To understand this trend and successfully conserve threatened species, population managers will require greater insight into

Corresponding author
Dominique A. Potvin,
dpotvin@usc.edu.au

population sex ratios, and how key drivers such as breeding systems and sex allocation may respond at the individual and population level (*Undin & Castro, 2022*). For example, even small sex skew variations from parity adversely affect *ex-situ* flamingo (*Phoenicopterus spp.*) populations, not only by limiting the number of breeding pairs, but *via* nest failures due to the disruptive behaviour of unpaired birds (*Mooney et al., 2023*).

Avian sex ratios may be calculated at various stages of development *i.e.*, at fertilisation (primary sex ratio), at hatch (secondary sex ratio) or as the adult sex ratio (tertiary sex ratio). The critical parameter when investigating the effects and conditions of skewed sex ratios, however, is the effective or operational sex ratio (OSR). The OSR is defined as the number of reproductively active males to females in a population, thus providing the most accurate assessment of population viability through reproductive output. Most commonly, the OSR is observed at the individual level, where one female may produce more sons or daughters, but can also be calculated at the population level.

Avian sex ratios rarely occur at parity for many physiological and environmental reasons (*Donald, 2007*). Regardless of the cause, sex skews may reduce the effective population size *via* an imbalance in the number of males or females available for breeding, which is a critical parameter of population growth, sustainability and adaptive potential in wildlife (*Frankham, 1995*). When the effective population size becomes too low, there may be a trend towards increased homozygosity through inbreeding, loss of genetic variability and consequent genetic drift (*Frankham, 1995*; *Charlesworth, 2009*). In addition to genetic factors, skewed sex ratios may also have severe effects on population dynamics and social behaviour of individuals in small or closed populations (*Tschumi et al., 2019*) and may exacerbate population declines and undermine conservation efforts. Observations of the causative factors of sex ratio variation can be made at various developmental levels; however, ultimately start with the primary sex ratio.

The physiological drivers behind avian sex allocation at the primary sex ratio level lay with the female. In birds, females are the heterogametic sex, carrying both a Z and a W sex chromosome (males ZZ), so theoretically, sex allocation is under maternal control (*Alonso-Alvarez, 2006*; *Dijkstra et al., 2010*; *Major & Smith, 2016*). The mechanisms for variation in subsequent life stages are usually due to sex-biased mortality ((*Potvin & MacDougall-Shackleton, 2010*; *Székely et al., 2014*); also see reviews by *Donald (2007)*, *Payevsky (2021)*).

Although a skewed sex ratio at any developmental stage may create short-term, proximate complications at the individual level, fundamental ecological theories suggest it will even out when viewed at the population level (*Trivers & Willard, 1973*; *Frank, 1990*). The precise mechanisms for this are unknown; however, some may correlate with species' mating strategies (*Nunney, 1993*; *Charlesworth, 2009*). Polygamous species are expected to exhibit a stronger tendency towards population sex skews compared to monogamous species, with population fitness decreasing more rapidly in monogamous species than in species exhibiting polygamy due to the limitation of mates (*Emlen & Oring, 1977*; *Frankham, 1995*). In addition to sex-specific mortality, sex ratio bias has also been attributed to numerous other causative factors, including sex-specific within-clutch laying orders (*Alonso-Alvarez, 2006*), or the size (*Anderson et al., 1993b*) or rate of maturation of

offspring (*i.e.*, either the slower to mature or those with the best chance of reproducing in the same season are laid first) (*Krebs et al., 2002*; *Dijkstra et al., 2010*). Other influences include the attractiveness of a mate (*Burley, 1981*), weather patterns (*Brooke et al., 2012*), causative factors of declining threat status (*i.e.*, small population size, alee effects) (*Donald, 2007*; *Morrison et al., 2016*) and deteriorating maternal condition throughout the laying period (*Nager et al., 1999*). If a predictable hypothesis for modelling sex allocation theory in birds is to be formed, it will require looking beyond theoretical studies of adaptive significance to include trends in genetics, ontogeny and phylogeny (*Frank, 1990*), and advancing our knowledge of the mechanisms of avian sex determination, species-specific mating strategies and various factors of an individual's multidimensional environment. Data from *ex-situ* populations may inform the conservation management of species in the wild and in zoo settings.

The *ex-situ* environment removes many hurdles of observing sex ratios and sex allocation in a wild setting by enabling direct access to offspring and life history data at all stages of development. Additionally, it can provide insight from longitudinal management and expertise at a global level to align theories and predict patterns that may be extended to related taxa (*Sheldon, 1998*; *Fisher & Owens, 2004*; *Mooney et al., 2023*). These data may also be incorporated into population models, inform experimental manipulations and highlight potential problems for conservation programs (*Taylor & Parkin, 2008*; *Conde et al., 2013*). The success of *ex-situ* programs; however, is limited by multiple factors including institutional participation (*i.e.*, the number of enclosures available), knowledge of species biology, and importantly, population sex ratios. Published accounts of avian *ex-situ* populations at genus and species level regularly report a sex skew to male (*Taylor & Parkin, 2008*; *Ferrie et al., 2013*; *Stojanovic et al., 2018*; *Mooney et al., 2023*); however this pattern may be shifting to more even sex ratios according to recent global analyses on avian offspring in *ex-situ* holdings (*Machado & Miller, 2023*).

If we are to effectively integrate the management of species between *in-* and *ex-situ* environments, understanding broad patterns of sex ratios in captive populations will increase management options through informing whether adaptive alterations to both environmental and social factors that influence sex allocation and fine-scale conditions during critical stages of ontogeny need to be considered (*Wedekind, 2012*). The ability to augment sex ratios will facilitate the strategic supplementation of additional males or females (*Lenz, Jacob & Wedekind, 2007*; *Wedekind, 2012*) to wild populations as required in threatened species recovery programs such as the orange-bellied parrot (*Neophema chrysogaster*) (*Stojanovic et al., 2018*; *Troy & Lawrence, 2021*), regent honeyeater (*Anthochaera phrygia*) (*Heinsohn et al., 2022*) and Andean condor (*Vultur gryphus*) (*Lambertucci et al., 2013*).

The conservation management of threatened species requires a multi-disciplinary approach based on a foundation of accurate, accessible data to inform adaptive management processes (*Conde et al., 2013*; *Schwartz et al., 2017*). The Zoological Information Management System (ZIMS) (*Species360, 2023*) is a sophisticated, real-time global records database of 22,000 animal species under human care and has been collecting data since 1974. More than 1,300 registered zoos and aquaria from 102 countries use the
system to manage data on species' life histories including births, deaths, parentage, movements, health and reproductive history. These data are shared globally by registered ZIMS members to achieve best practice in animal management and conservation goals (*Species360, 2023*). The increasing use, and success, of integrated management strategies (*Schwartz et al., 2017*) have seen the expansion of the database to link information about animals that transition between *in-situ* (in the wild) and *ex-situ* (in zoological facilities) environments. The mass of data collected on these species to date forms a rich resource for conservation research, including areas of reproductive biology, life history, ontogenesis, diagnosis and treatment of illness, nutrition, behaviour and the formation of biomaterial banks. Reports generated from ZIMS data present an unparalleled opportunity to understand animal reproductive biology in a controlled setting without the experimental manipulation of threatened species (*Farquharson, Hogg & Grueber, 2021*). Despite the quality and accessibility of this data, there is very little existing literature using this resource to observe sex ratio variation in birds on a global scale (*Machado & Miller, 2023*).

Using ZIMS data, we investigated the prevalence of sex skews in global avian populations. We used quantitative modes of enquiry to investigate the global proportion of males in 30 orders of birds. Five of these orders were chosen to examine in greater detail how the proportion of males related to (i) IUCN threat status, (ii) broad-level phylogenetic relationships and (iii) how the proportion changed at multiple scales (institutional to global). Three of these five focus orders were further selected to investigate the effect of housing within or outside of their natural range on the proportion of males. The findings in this study provide the first broadscale examination of *ex-situ* sex ratios and will contribute to the field of integrated species management.

## MATERIALS AND METHODS

To examine global patterns in sex ratios, all current (as at 01/03/2023) *ex-situ* avian holding records were exported from the ZIMS global resources portal, excluding domestics, breeds and varieties. All ambiguous taxonomy (*i.e.*, listed only at genus level, or listed too broadly, such as 'ducks, geese and swans') and obsolete taxonomy (*i.e.*, historical data from superseded taxonomic classification, as compared with the Birds of the World list as of the date 23/03/2023) were removed. Data listings were also removed if there were no known males or females (*i.e.*, *all* individuals listed as unknown sex) or if there was only a single institution listing for that species. This resulted in a final dataset with 30 orders from 39,964 institutional listings. All subsequent analyses used institutional listings as a sample; herein described as a holding. Data regarding the holdings were limited to species (as well as genus and order), IUCN listing, holding mnemonic, holding location, number of males, number of females and number of 'other' (unknown) sex. Any further detail regarding the origin (*e.g.*, wild-caught, captive-bred) of the individuals in the holding or the type of housing (*e.g.*, indoor, outdoor aviaries *etc.*) were not included for this broadscale analysis. It is likely that most institutions reported the sexes of individuals at maturation, and that this study consequently examines adult sex ratios.

The sex ratio was calculated for each holding as a proportion of males to females, *i.e.*, (M/(M+F)). These proportions were then used to calculate the mean global sex ratio

(*i.e.*, from all holdings within each of the 30 orders) as an exploration of the data. All subsequent analyses, however, utilised the full dataset using each institutional holding as an individual sample.

For the full dataset, significant differences in sex ratios (using each holding as an individual sample) were tested between (a) orders and (b) genera, to provide a broad indication of taxonomic signature on sex skews. That is, we used sex ratio (presented as a decimal indicating the proportion of males) of each holding as a response variable, and (a) order, and (b) genus as predictor variables to understand if sex ratios differed between these higher taxonomic groups. Jamovi version 2.2.5.0 (*The jamovi project, 2023*) was used for all inferential data analyses. Data were checked for skewness and normality (Shapiro-Wilk), with parametric or non-parametric equivalents applied where appropriate. A two-way analysis of variance (ANOVA) with Tukey's *post-hoc* pairwise comparison with Bonferroni correction was employed to examine whether significant differences in the proportion of males were present between the predictor variables of taxonomic grouping (genus and order). Five focal orders were chosen for additional statistical analysis due to the quality of data, coverage of a wide variety of species, and global holding trends (*i.e.*, 'popular' groups). They included the Struthioniformes (ratites), Psittaciformes (parrots and cockatoos), Columbiformes (pigeons and doves), Passeriformes (perching birds) and Coraciiformes (Kingfishers and allies). For these analyses, threat categories were included as a predictor variable, as set out by the International Union for the Conservation of Nature (*IUCN, 2023*) and included least concern, near threatened, vulnerable, endangered, critically endangered and extinct in the wild. To determine the effects of housing a species within or outside of their natural range, *in-situ* distribution as listed by the *IUCN (2023)* was compared to the country of *ex-situ* holding (ZIMS) in the Struthioniformes, Columbiformes and Psittaciformes. These orders were chosen due to their broad global distribution within the database and the presence of previously identified sex skews.

As with the previous analyses, data were checked for skewness and normality, with parametric or non-parametric equivalents applied when appropriate. A two-way ANOVA with Tukey's *post-hoc* pairwise comparison were employed to examine whether differences in the proportion of males were present between IUCN threat status categories. An independent t-test was used to examine if the proportion of males significantly differed when holdings were inside or outside a species' natural range.

To confirm the validity of our results, we calculated global sex ratios for (a) all orders and (b) all genera using a dataset which only included species with 100 holdings or more. We then compared these to the full dataset, to examine whether small token holdings were contributing to any observed skews in the above analyses.

Finally, in order to account for phylogenetic relationships and avoid pseudoreplication, we performed a Bayesian mixed model in R (*R Core Team, 2021*) using the package brms (*Bürkner, 2017*). First, we created a phylogenetic tree using our dataset through birdtree. org (*Jetz et al., 2012*), using the Ericson All Species source (10,000 trees with 9,993 operational taxonomic units (OTUs) each), and selecting a maximum likelihood tree using the phangorn package (*Schliep, 2011*). We then ran the mixed model, testing IUCN status and holding institution effects on the proportion of males at a species level, with the

phylogenetic random effect included. We used uninformative priors, two chains, a warmup of 1,000 runs and an iteration of 3,000 runs for this analysis.

For the purposes of this study, populations were considered skewed if they deviated from the expected Mendelian sex ratio of 1:1 (*Fisher, 1930*). Given that observed proportions rarely conform to exact parity due to natural variation and dataset limitations, we applied a buffer of ±0.005 to account for minor fluctuations (*Donald, 2007*). Populations were categorized as male-skewed if the proportion of males was ≥0.505 and female-skewed if the proportion of males was ≤0.494. ZIMS data were current as of 23 March 2023.

# RESULTS

## Proportion of males in global avian orders and genera

Overall, we found that 61% of all genera and 80% of all orders were skewed to male, 13% of genera and 16% of orders were skewed to female and 7% of orders and 8% of genera were at parity (Fig. 1 and Table S1). In the validation stage, where only species with 100 or more institutional holdings were analysed, percentages were found to be more even with genera consisting of 69% male skewed, 17% female skewed and 13% at parity; and orders of 68% male skewed, 16% female skewed and 8% at parity. Our ANOVA results showed that taxonomic rank affected the proportion of males at the genus level ($p < 0.001$); however, not at the level of order ($p > 0.05$) (Table 1). Orders that were skewed to female also showed greater variability between genera.

## Effect of threat status on proportion of males

Of the five focus orders, ANOVA results only revealed a significant variation in the proportion of males within IUCN threat categories in the Columbiformes $F(5,1999) = 2.36$, $p = 0.038$ (Table 2). *Post-hoc* testing showed that the variation within the order Columbiformes was driven by a significant difference between the vulnerable ($M = 0.53$, $n = 218$) and extinct in the wild ($M = 0.70$, $n = 40$) categories $p = 0.013$ (Table S2).

In our analysis that accounted for phylogenetic relationships, our results were similar, suggesting our overall patterns were robust. No strong evidence of an effect of IUCN listing category on the proportion of males was found. Estimates were as follows: Intercept or reference category (Data deficient) 0.57 ± 0.04 (95% CI [0.49–0.65]); Critically Endangered (0.00 ± 0.04, 95% CI [−0.08 to 0.08]); Endangered (−0.05 ± 0.04, 95% CI [−0.13 to 0.03]); Extinct in the Wild (0.08 ± 0.06, 95% CI [−0.04 to 0.19]); Least Concern (−0.04 ± 0.04, 95% CI [−0.12 to 0.04]); Near Threatened (−0.05 ± 0.04, 95% CI [−0.13 to 0.03]), and Vulnerable (−0.05 ± 0.04, 95% CI [−0.13 to 0.03]). As can be seen, all 95% credible intervals included zero, suggesting no significant effect.

Random effect variance estimates from the Bayesian model also indicated minimal phylogenetic signal ($\sigma = 0.01$) at the species level, while population-level variation (*i.e., holding*) was slightly higher ($\sigma = 0.02$). This, again, is congruent with our initial analyses above. Model convergence was confirmed ($\hat{R} \approx 1.00$ for all parameters), and effective sample sizes were sufficient (Bulk_ESS > 1,500).

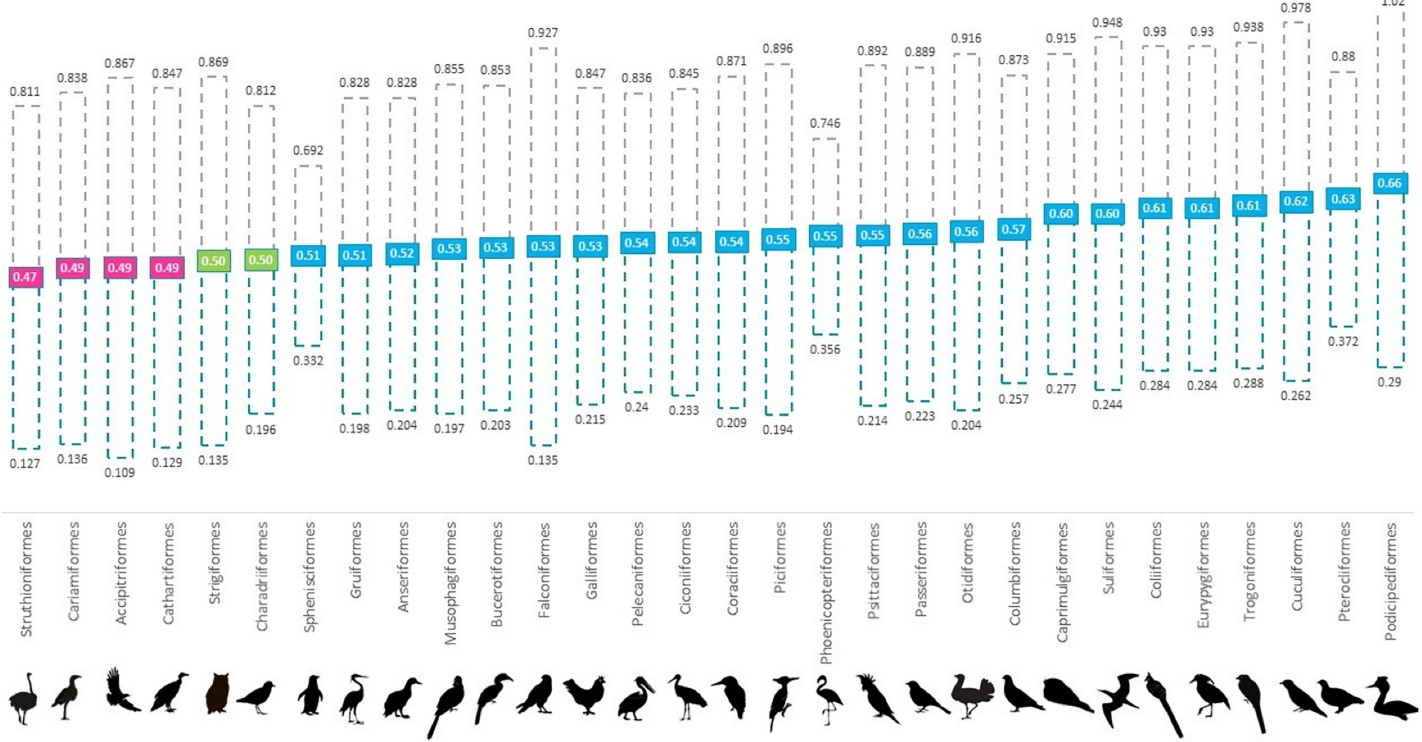

**Figure 1 Mean proportion of males for 30 avian orders per holding in global *ex-situ* collections showing spread from 35–50 percentile to 50–65 percentile.** Orders are listed in increasing order of proportion of males from left (lowest) to right (highest). Pink boxes indicate a skew to female (<0.494), green indicate at parity (0.495–0.504) and blue boxes indicate a skew to male (>0.505).

**Table 1 ANOVA test results showing the effect of, and interaction between, taxonomic grouping (genus and order) on the proportion of males in the global holdings of 30 avian orders.**

| Source | df | F | Sig. |
|---|---|---|---|
| Corrected model | 829 | 1.983 | <0.001 |
| Intercept | 1 | 4,507.427 | <0.001 |
| Order | 2 | 0 | 1 |
| Genus | 774 | 1.798 | *<0.001 |
| Order * Genus | 26 | 0.021 | 1 |

Notes:
a. R squared = 0.040 (Adjusted R squared = 0.020).
Asterisk indicates significant result. *df*, degrees of freedom; *F*, F statistic.

## Proportion of males in holdings inside and outside of natural range

Only 5% of Struthioniformes, 11% of Psittaciformes and 12% of Columbiformes are housed within their natural range. There were no significant differences in the proportion of males held within or outside of their natural range (all $p > 0.05$); however, there is a trend that existing sex skews are modestly accentuated outside of a species' natural range (OR) compared to inside the natural range (IR), for example, in the female skewed Struthioniformes (OR $M = 0.46$, IR $M = 0.50$) and male skewed Psittaciformes (OR $M = 0.55$, IR $M = 0.54$) and Columbiformes (OR $M = 0.57$, IR $M = 0.55$) (Table 3).

**Table 2 ANOVA results showing the variation in proportion of males among orders within IUCN threat categories.**

| Order | Mean, SD n | df | F | p |
|---|---|---|---|---|
| Psittaciformes | 0.553, 0.339 6,621 | 6 | 1.42 | 0.201 |
| Struthioniformes | 0.47, 0.343 1,210 | 2 | 1.30 | 0.278 |
| Columbiformes | 0.565, 0.308 2,005 | 5 | 2.36 | *0.038 |
| Passeriformes | 0.556, 0.333 4,323 | 9 | 0.58 | 0.815 |
| Coraciiformes | 0.540, 0.330 636 | 2 | 0.571 | 0.565 |

Note:
Asterisk indicates significant result. *n*, sample size; *df*, degrees of freedom; *F*, *F* statistic.

**Table 3 Descriptive statistics showing the mean proportion of males by order, when housed within or outside of their natural range.**

| Order | Within range Yes/No | n | Mean |
|---|---|---|---|
| Struthioniformes | Yes | 61 | 0.50 |
| | No | 1,141 | 0.47 |
| Columbiformes | Yes | 238 | 0.55 |
| | No | 1,767 | 0.57 |
| Psittaciformes | Yes | 727 | 0.55 |
| | No | 5,894 | 0.55 |

Note:
Struthioniformes data is accentuated in the opposite direction due to an overall mean skew to female (*i.e.*, decreased proportion of males when held outside of range). *n*, sample size.

## DISCUSSION

An initial objective of this research was to observe sex skews and avian sex ratio variation in global *ex-situ* populations. With respect to this, we found that although *ex-situ* sex ratios varied significantly at the institutional level, the orders were closer to parity at a global scale. This was a consistent factor throughout the 30 orders. Most striking was the substantial and unexpected finding in the disparity between the large proportion of orders with sex ratios skewed to male compared to female. Although there is evidence that because female Aves are heterogametic, there is a tendency for sex ratios to be male-biased (*Pipoly et al., 2015*), it is interesting to see the variability between groups within the class. The proportion of males held in each order consistently showed statistically significant variability, which in large-bodied charismatic avian families, such as some raptors (Accipitriformes, Falconiformes and Strigiformes) and parrots (Psittaciformes) should be interpreted with caution as members of these orders are also held singly for their sexually dimorphic characteristics, free-flight shows or other in-house education or entertainment purposes (*Wilkinson, 2000*; *Brereton & Brereton, 2020*), rather than with an intention to breed (*Frynta et al., 2010*). This is obviously an important factor in understanding the

causative factors of *ex-situ* sex skews as it may explain, for example, variations in annual reproduction. Very little was found in the literature to further analyse and compare our results of broadscale sex ratio variation in *ex-situ* avian populations, which is surprising given the abundance and quality of data available. However, we can look to studies of wild birds, and accounts of individual species both *in-* and *ex-situ* to give our findings context and advance our understanding of sex ratio variation in *ex-situ* avian populations.

In sequence of increasing proportion of males, the basal avian order Struthioniformes (ratites) had the lowest mean proportion of males, followed by four taxonomically linked orders under the raptorial banner—the Cariamiformes (seriemas), Accipitriformes (hawks, eagles, vultures and kites), Cathartiformes (new world vultures) and Strigiformes (owls) (*Jarvis et al., 2014*; *McClure et al., 2019*; *Méndez et al., 2022*). A notable omission in this sequence is the taxonomically separated raptor order Falconiformes (falcons and caracaras), which is skewed to male. There are similarities between our broadscale findings and published accounts (*Clutton-Brock, 1986*; *Xirouchakis et al., 2023*). The majority of raptor orders exhibit reverse sexual size dimorphism whereby females are heavier than males (*Massemin, Korpimäki & Wiehn, 2000*; *McDonald et al., 2005*) so their skew to female contravenes empirical sex allocation theories. This direction of population sex skew to the larger sex also accords with an order with perhaps the most extreme example of sexual size dimorphism in the Aves, although this time to males, the bustards (Otidiformes).

*Ingraldi (2005)* postulated that a male bias in the usually female-skewed Accipitriformes was due to environmental stress. This is interesting as the release of the stress hormone corticosterone in the maternal female can trigger brood sex ratio adjustment and bias offspring sex ratios to male (*Schoech et al., 2009*; *Geffroy & Douhard, 2019*) as also reported in Passeriformes (*Gam, Mendonca & Navara, 2011*; *Pryke et al., 2011*) and Galliformes (*Pinson et al., 2011*). In a captive setting, genotypes and phenotypes of less stressed individuals may be selected for, due to greater fecundity and lower mortality over consecutive generations (*Price, 1999*; *Crates, Stojanovic & Heinsohn, 2023*). Such changes may be subtle and add to the complexity of understanding the phenotypic costs of captivity and contradictions of the Fisherian theory of sex allocation such as female-skewed ratios in raptors.

In addition to links in sexual size dimorphism, a difference in maturation age (age at first breeding) is also apparent in the raptorial orders (*Daan, Dijkstra & Weissing, 1996*) and bustards (Otidiformes) (*Alonso et al., 2010*), which have pervasive implications in sex allocation theory. For example, in raptors, female biased broods may occur early in the season as they reach sexual maturity earlier than males and thus have a better chance of reproducing first (*Alonso et al., 2010*; *Carmona-Isunza et al., 2017*) and the converse in species where males have a faster maturation rate (*Daan, Dijkstra & Weissing, 1996*). In long-term studies of raptor sex ratios, it was found that although brood sex ratios changed throughout each season (*i.e.*, female biased in early and first hatched chicks and male biased in late and third hatched chicks or vice versa) the sex ratio at population level was closer to parity over time (*Dijkstra, Daan & Buker, 1990*; *Daan, Dijkstra & Weissing, 1996*; *Xirouchakis et al., 2023*). This highlights the importance of communicating the timing and

type of sex ratio in the literature, and the use of this knowledge in *ex-situ* or integrated management programs where only first eggs or clutches may be permitted to hatch per season as a method of limiting production. It is especially relevant for conservation programs of critically endangered species such as the Bengal florican (*Houbaropsis bengalensis*) which has the added complexity of requiring a minimum captive release to the wild ratio of at least four males to one female for the lek mating strategy to function (*Mahood et al., 2021*).

Other studies that have found sex biases in captive populations have noted that supplementary feeding of female birds may encourage an excess of male offspring, leading to male biased sex ratios (*Clout, Elliott & Robertson, 2002*). This is one explanation put forward for a more limited study examining 80 species of Psittaciformes that found an overall male bias in captive holdings (>70% of species; *Taylor & Parkin, 2008*). Our study found an overall skew to males in Psittaciformes, with a mean sex ratio of 0.55, which was not extreme, and indeed individual holdings showed a wide range of sex ratios, indicating an intermediate result compared with this previous study and that from *Machado & Miller (2023)*, who found that offspring sex ratios of Psittaciformes have historically been male-biased, but since 2000 demonstrate a median ratio of 50:50. This pattern of heavier biases in earlier time periods appeared to be common across many taxonomic groups (*Machado & Miller, 2023*). We did not delineate between juveniles and adults, nor did we make observations on offspring sex ratios specifically, which may account for our findings showing a moderate sex ratio bias in many taxa, not as skewed as historical datasets but also not predominantly at parity either, as it appears many taxa have become in captive settings (*Taylor & Parkin, 2008*; *Machado & Miller, 2023*).

Some of the species demonstrating male bias are cooperative nesting species that engage nest helpers, which is also supported in the available literature (*Emlen, Emlen & Levin, 1986*). This trend supports Hamilton's kinship theory (*Hamilton, 1967*), that is, that parents will invest more in the helping sex where nest helper investment increases with kinship (*Brown, 1978*; *Frank, 1990*; *Green, Freckleton & Hatchwell, 2016*). Parallel to this, is the repayment model where the helping sex repays the breeding pair by way of helping with consecutive broods, and thus become the cheaper sex and support Fisher's theory of equal investment (*Emlen, Emlen & Levin, 1986*). This may explain the skews to the helping sex in numerous genera, for example the Coraciiformes (kingfishers and allies) genera Halcyon ($M = 0.75$) and Merops ($M = 0.55$), and Passeriformes (perching birds) genera of Malurus ($M = 0.75$) and Picoides ($M = 0.67$). This is challenging in a zoo setting as family groups are infrequently kept together post breeding season, possibly further exacerbating the outcome of a male skewed ratio.

## Threat status

The second objective of this study was to observe if the proportion of males varied between IUCN threat categories in our five focus orders, as *in-situ* studies have noted a positive correlation between rising threat status and sex skews (*Donald, 2007*; *Morrison et al., 2016*; *Payevsky, 2021*). However, the findings of this *ex-situ* study did not overwhelmingly support the previous research. Of the five focus orders, we only found a significant

variation between the proportion of males between Vulnerable and Extinct in the Wild threat categories in the Columbiformes (pigeons and doves). Less than 15% of the total *ex-situ* holdings of pigeons and doves are listed at an IUCN threat level of Vulnerable or above, with the vast majority not classified as threatened (*i.e.*, Least Concern and Near Threatened). The interpretation of these results would be improved with the additional independent variable of management type (*i.e.*, by institution, regional and/or global studbook or none) as an indication of regional or global species management commitment and focus. This brings to light an important issue for global collections, that is, how to find a defensible balance between holding charismatic species and those requiring intensive conservation commitment.

## Outside of range

Global partners can play an important role in expanding the holding capacity and husbandry experience for a species; however, our study has revealed it may also create unforeseen problems. We found that skews, although not at a statistically significant level, were generally more pronounced in regions outside a species' natural range. In small or isolated populations, this small increase in population skew may still cause deleterious temporal effects. It is well known that large, brightly coloured, and active birds are important in zoo collections to lure visitors and help create defining moments and experiences (*Frynta et al., 2010*). Within the global avifauna, a disproportionate number of these are endemic to tropical regions where breeding season is intimately linked to rising barometric pressure from developing low pressure systems (*Cooney et al., 2022*). However, our data show that over 50% of the global holdings are in Europe where the temperate weather patterns are dissimilar to the tropics, with generally dry summers and cloudy to wet winters. Although many of these species may be held in climate-controlled enclosures to protect from cold temperatures, the difference in atmospheric pressure may still trigger winter breeding.

On a physiological level, this difference in climatic region presents numerous opportunities for conflicting messages within critical regions of the brain and gonads responsible for essential physiological changes at the onset of a breeding cycle (*Gwinner, 2003*; *Leska & Dusza, 2007*). The outcome in our study of a skew to male in tropical species held in temperate climatic regions, broadly supports the sex allocation theory of local resource competition, where the low barometric pressure of the Northern hemisphere summer may signal a food shortage in the coming season and skew the sex of offspring to favour the natal dispersing sex (*Gowaty, 1993*; *Weatherhead & Montgomerie, 1995*). For example, we found the skew to male in Cuculiformes (cuckoos) was largely driven by two genera with 100% of holdings in temperate regions with dry summers in contrast to their tropical *in-situ* summer range (*Payne & Sorensen, 2005*; *Cariello, Macedo & Schwabl, 2006*). Consistent with these factors, there was also modest evidence that holding members of the Psittaciformes (parrots and cockatoos), Columbiformes (pigeons and doves) and Struthioniformes (ratites) outside of their range exacerbated sex skews in the population, although for the ratites it occurred in the opposite direction towards females.

A final consideration proposed by *Cariello, Macedo & Schwabl (2006)* is that the shorter the breeding season, the higher the testosterone level in tropical birds. Circulating hormones in the maternal body are a known manipulator of avian sex determination, which may be an additional driver when tropical species are exposed to shorter breeding cycles at higher latitudes. It would give clarity to these proposals to compare sex skew variation across weather patterns, as well as region.

## Differences in *ex-situ* and *in-situ* environments

The differences between housing birds *in-* and *ex-situ* are immense. For example, wild populations experience sex differences in the effects of survival, dispersal and temporal and spatial abundance (*Morrison et al., 2016*); however, we can see from our data that the direction and presence of sex skews in avian orders are mostly similar to patterns in other studies that have been observed in wild populations (*Donald, 2007*; *Payevsky, 2021*). Two comprehensive reviews of sex ratios in wild avian populations (*Donald, 2007*; *Payevsky, 2021*) found that approximately one third of all published estimates were skewed to male (33% and 16–30%) from 200 to 308 records respectively. This trend was magnified in our study where we found 61% of all genera and 80% of all orders were skewed to male. When this calculation was modified to include the more robust sample size of no less than 100 individual holdings, the male skew was more even between the taxonomic levels, at 69% of genera and 68% of orders. Although striking, this result may be partly explained by the following factors. Firstly, *Payevsky (2021)* notes that sex ratios start to be male-biased at fledge. In an *ex-situ* setting, the potential of sex-biased mortality from environmental or social factors is minimised, meaning all chicks theoretically have a more equal chance to survive to maturity and beyond. Secondly, with few exceptions, *ex-situ* populations are closed systems where birds are unable to disperse or be managed out of the global zoo system, and thirdly, a small but consistent contributing factor may be that a higher sex-specific mortality rate has been shown to occur in the heterogametic sex of a variety of taxa including birds (*Clutton-Brock, 1986*; *Liker & Székely, 2005*; *Donald, 2007*; *Xirocostas, Everingham & Moles, 2020*; *Payevsky, 2021*). Despite this, it is unclear how contrasting sex determination practices (*e.g.*, using genetics, morphometrics or other methods, at juveile or adult stages) might result in differences between institutions in their timeliness or ability to report sex ratios, nor do we know whether other practices such as removing eggs or chicks for hand-rearing might either inadvertently or purposely result in skewed sex ratios. This in and of itself highlights the importance of considering universal methods or standards for sexing individuals in *ex-situ* holdings.

*In-situ* research shows that secondary sex ratios are usually close to parity, and it is post-fledge where sex ratios become biased (*Clutton-Brock, 1986*; *Payevsky, 2021*; *Wood et al., 2021*). In some threatened species, such as the orange-bellied parrot, this has intensified with the survival of juveniles in their first year now less than half that of previous decades (*Stojanovic et al., 2020*). It is apparent that the main driver of wild population level adult sex ratios rests with sex specific mortality and that future research would benefit from concentrating on adult sex ratio rather than the already abundant literature on primary and secondary sex ratio influences. With respect to *ex-situ*

populations, where survival is likely more equal between the sexes, this points more to a research focus of primary and secondary sex ratios. This is intuitive as we now have a better understanding of the potential of maternal sex determination and the effects of environmental stressors (*Schoech et al., 2009*; *Geffroy & Douhard, 2019*), hormones (*Smith, Katz & Sinclair, 2003*; *Major & Smith, 2016*) and endocrine disrupting substances in the maternal environment (*Colborn, Saal & Soto, 1993*; *Piferrer & Anastasiadi, 2021*).

In broad support of the Trivers-Willard hypothesis (*Trivers & Willard, 1973*), females of most avian species when in good condition will produce more males. In an *ex-situ* setting with a constant abundance of available food, this opportunity may occur more often than for wild counterparts. As well as producing a skew to the expensive sex, an abundance of food may also interfere with the sex ratio of offspring at fledge, or, the paradox of hatching asynchrony where the hatching order within a clutch directly contributes to the often sex-specific death of later-hatched young (*Magrath, 1990*). This possibility is supported by a review of *in-situ* sex skew data by *Payevsky (2021)* who found that sex ratios start to show a male bias at fledge. An abundance of food may reduce occasion for sibling competition and siblicide (*Bortolotti, 1986*; *Drummond et al., 1991*) the effect of natal sexual size dimorphisms (*Anderson et al., 1993a*; *Benito & González-Solís, 2007*) and promote the survival of all chicks regardless of sex or fitness. Lastly, *Kappeler et al. (2023)* found a positive correlation between a species' sensitivity to sex skews and the increasing diversity of the community within which they live, even over short periods of time. The very nature of a modern zoo is that multiple species may be held in close proximity to each other, whether that be directly within the same enclosure, or in visual or auditory contact.

## Sex skew management *ex-situ*

The methods institutions or programs use to manage an excess of males or females depends on numerous practical factors including holding capacity (*i.e.*, number of available enclosures), inter- and intra-species compatibility, access to reproductive technologies such as artificial insemination (*Blanco et al., 2009*; *Rodger & Clulow, 2022*), in-ovo sexing (*Jensen, Mace & Durrant, 2012*; *Hall, Potvin & Conroy, 2023*) and in a more specialised context, the xenogenic transfer of avian germplasm into a host embryo (*Roe et al., 2013*; *Imus et al., 2014*), although the full potential of this is yet to be realised as a conservation management option (*Mastromonaco & Songsasen, 2020*). Other solutions include improved cooperation between regional and international institutions and the exchange of individuals between *in-* and *ex-situ* populations (*Keulartz, 2023*).

Despite the many options for managing sex ratio variation both *in-* and *ex-situ*, there were no avian examples found in the literature of ways to proactively manage sex skews or manage the sex of offspring prior to lay in a zoo setting (although see *Wedekind, 2002, 2012*). It is worth noting, that an abundance of literature exists for manipulating the primary and secondary sex of poultry (*Abinawanto & Saito, 1997*; *Vaillant et al., 2001*; *Smith, Katz & Sinclair, 2003*) however these methods have not been adopted in a conservation setting. If species are to be managed integratively, we need to improve reproductive technologies to promote the increase of populations in a structured way without losing genetic diversity (*Wildt & Wemmer, 1999*). The diametric challenge is to

prevent the hatching of birds of the undesired sex so as to not take up the limited resources available in most zoos, or theoretically in wild populations for example, the endangered southern cassowary (*Casuarius casuarius johnsonii*), where to improve low fecundity rates, institutions are requested to house two males per breeding female to allow for mate choice. Therefore, the ability to match this in reproductive output would theoretically improve reproductive outcomes for the species (*Hall, Potvin & Conroy, 2023*). The current limits of our practical and theoretical knowledge are ultimately preventing the application of such solutions.

## CONCLUSIONS

The purpose of this study was to gain a better understanding of sex ratio variation in global avian populations. The results show that sex skews to male are more than twice as prevalent in *ex-situ* populations than with their wild counterparts, and although many parallels were found within our data, the level of variation was often inconsistent with mainstream sex allocation theory. The findings of this investigation do, however, broadly support the Trivers-Willard hypothesis that breeding females will produce the sex which will return the highest reproductive value, which for female birds in good condition is usually male. Taken together, these findings have significant implications for the understanding and management of sex skew variation in *ex-situ* populations, especially for threatened species and integrated systems of management. Further research incorporating annual births/hatch data, as well as the motivation behind collection holding numbers from institutions would be useful. Additionally, investigations should be undertaken to examine links between primary sex ratio (PSR) and environmental stressors, maternal corticosterone and diet in a conservation or captive breeding context. To develop a better understanding of the social environment, exploring the effects of removing offspring on the sex of consecutive clutches in philopatric, cooperatively breeding species would be beneficial. Future studies are also recommended on the effects of barometric pressure and housing tropical species in temperate climatic regions. Overall, there is abundant room for progress in understanding sex allocation theory in birds, how it differs between taxonomic groups, and how it responds to change in both *in-situ* and *ex-situ* environments such that integrated management programs might have the best conservation outcomes.

### Funding
The authors received no funding for this work.

### Competing Interests
The authors declare that they have no competing interests.

### Author Contributions
- Clancy A. Hall conceived and designed the experiments, performed the experiments, analyzed the data, prepared figures and/or tables, and approved the final draft.

- Gabriel C. Conroy conceived and designed the experiments, authored or reviewed drafts of the article, and approved the final draft.
- Dominique A. Potvin conceived and designed the experiments, authored or reviewed drafts of the article, and approved the final draft.

## Data Availability

The raw data is available in the Supplemental Files.

## Supplemental Information

Supplemental information for this article can be found online at http://dx.doi.org/10.7717/peerj.19312#supplemental-information.

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
