# Peer review of "Ex-situ avian sex skews: determinants and implications for conservation"

_PeerJ, doi:10.7717/peerj.19312_

## Round 0.1 · original submission · Major Revisions

As you can see, the three reviewers have made detailed suggestions for revising your manuscript. In this we are lucky. There is nothing worse that getting a short, uninformative review with the recommendation to reject. So, please work through these comments, respond to each, and send me your revision.

Reviewer 1 ·

Basic reporting

1. [150] How was official taxonomy determined and what taxons were considered obsolete/ synonymous? Avian taxonomy changes yearly (at the very least) due to new genetic data and forms of analyses. These changes are announced at the International Ornithological Congress, which runs the World Bird List. The list itself changes within the year. ZIMS data is extremely disorganized, with many synonymous species names active. For example, in your dataset you include Amazona lilacina and Amazona autumnalis as different species, however the World Bird List 14.2 (the most recent edition) considers Amazona lilacina to be a subspecies of Amazona autumnalis. There are likely several of repeats of this error throughout your dataset if up-to-date literature was not consulted.

2. [146-189] Your statistical procedure should be more detailed and specific. For example, throughout your multiple analyses it is not always clear when your outcome variable is sex ratio per holding, sex ratio per holding per order, sex ratio per holding per genus, etc. This is important to understand as IUCN and home range data can only be applied at the species or subspecies level. Similarly, you never specify what your final analysis was [183-189]. From reading your results I am inferring that you repeated the previous 3 ANOVAS and t-tests with a tougher exclusion criterion, but this should be clearly stated in the methods themselves. Additionally, on line [152] you states that data was removed if there was only 1 institutional listing, but it’s not clear if that’s per species, order, etc.

3. [109] A great deal of work has been conducted looking at avian sex ratio skews on a global scale, you mention some of such work later in your discussion (Donald, 2007; Payevsky, 2021, etc). [134-136; 143-144] You make a similar claim here, but sex ratios have been calculated in birds across the global ZIMS dataset as well (Machado & Miller, 2023). [231] It’s important to mention work which has been done looking at ex situ avian sex ratio skews and compare to their results (Nesterenko & Kashentseva, 2010; Machado & Miller 2023; & Taylor & Parkin, 2008).

4. [151-152] How many individuals were categorized as “unknown?” This could correlate with other important factors like age at mortality or husbandry practices and is important to know. Furthermore, in your discussion [sp 345-353] you should consider how differing sex practices between institutions could allow sex-biased juvenile mortality to skew sex ratios. IE if the animals are not sexed prior to becoming sexually dimorphic, they will show in the system as “unknown.”

5. [147-154] Were wild-caught or animals from unknown origins removed from the dataset? This is important information to share.

6. [148] What is considered a domestic and how was that determination made? ZIMS data ordered at the species name will combine listed subspecies together so that “Gallus gallus” includes both “domestic fancy” and “red jungle fowl.” Other issues can pop up around semi-domesticated breeds like swans, budgies, etc. It would be good to have a list of exactly who was excluded and at what taxonomic level.

7. [310-311] Both of these statements need citations.

8. Figure 1 is very well done and informative but might benefit from clarifying that it displays sex ratios per holding.

9. Overall your writing is clear and professional at the level of the sentence. The vast majority of statements contain references with sufficient background provided.

Donald, P. F. (2007). Adult sex ratios in wild bird populations. Ibis, 149(4), 671-692.
Nesterenko, O.& Kashentseva, T. Female-Biased Sex Ratio of Nestlings of Captive Red-Crowned Cranes in Russia. 2010. Proceedings of the VII European Crane Conference: Breeding, Resting, Migration and Biology, Stralsund, Germany. 123–125.
Machado, J. M., & Miller, L. J. (2023). Identifying Offspring Sex Ratio Skews in Zoological Facilities Using Large Historical Datasets. Journal of Zoological and Botanical Gardens, 4(4), 680-691.
Payevsky, V. A. (2021). Sex ratio and sex-specific survival in avian populations: A review. Biology Bulletin Reviews, 11(3), 317-327.
Taylor, T. D., & Parkin, D. T. (2008). Sex ratios observed in 80 species of parrots. Journal of Zoology, 276(1), 89-94.

Experimental design

1. [187-189] Deviations from a 50:50 sex ratio should be determined using an inferential statistical test, not using arbitrary criteria (Wilson & Hardy, 2002). You will also likely need to use a bonferroni correction.

2. Any kind of comparative analyses across taxonomic levels should be done with phylogenetic comparative statistics to avoid psuedoreplication. To put it simply: taxonomic orders are not independent. Species, genera, and orders are not equally related to other species, genera, and orders. Genetic information must be included to account for this. These readings may be helpful: (Mayhew & Pen, 2002; Mellor et al, 2018) Genetic information on most species can be obtained through weboflife.org and incorporated into your statistics fairly easily on R. There are also other platforms which specialize in phylogenetic comparative analyses (some of which are listed in Mayhew & Pen, 2002).

3. [155] Sex ratios calculated sensu stricto can lead to statistical and interpretation errors. They should instead be calculated as a proportion (M / M+ F). See Wilson & Hardy, 2002 for more information.

4. [147-154] You do not list a time cut-off in your exclusion criteria. ZIMS data goes back to the 19th century and many zoos have been around for over 100 years. Animal husbandry is a fast-changing field (Ward et al, 2018) and numerous variables like diet quality (Bradbury & Blakey, 1998; Rustein et al, 2004) and stereotypic behaviors (Martin et al, 2020) are known to correlate with skewed sex ratios. Thus, comparing a population from 1923 to 2023 is comparing apples to oranges. It may be worth considering introducing time periods to try and control for this issue.

5. [173] Your method of using country names to determine home-range is a bit clunky. In large countries such as the United States, these home ranges are vastly overexaggerated by this measure. For example, an arctic species like a Willow Ptarmigan would be considered “within home range” in a Miami zoo. A measure using longitudes might be more accurate and avoid errors like this. Furthermore, it stands a better chance of including introduced populations, some of which have been breeding outside of their native range for hundreds of years and are quite successful (example Canada Goose, European Starling, House Sparrow, etc).

Bradbury, R. B., & Blakey, J. K. (1998). Diet, maternal condition, and offspring sex ratio in the zebra finch, Poephila guttata. Proceedings of the Royal Society of London. Series B: Biological Sciences, 265(1399), 895-899.
Martin, M. S., Owen, M., Wintle, N. J., Zhang, G., Zhang, H., & Swaisgood, R. R. (2020). Stereotypic behaviour predicts reproductive performance and litter sex ratio in giant pandas. Scientific Reports, 10(1), 7263.
Mayhew, P. J., & Pen, I. (2002). Comparative analysis of sex ratios. Sex ratios: concepts and research methods. Cambridge Univ. Press, Cambridge, UK, 132-156.
Mellor, E., McDonald Kinkaid, H., & Mason, G. (2018). Phylogenetic comparative methods: Harnessing the power of species diversity to investigate welfare issues in captive wild animals. Zoo biology, 37(5), 369-388.
Ward, S. J., Sherwen, S., & Clark, F. E. (2018). Advances in applied zoo animal welfare science. Journal of Applied Animal Welfare Science, 21(sup1), 23-33.
Wilson, K., & Hardy, I. C. (2002). Statistical analysis of sex ratios: an introduction. Sex ratios: concepts and research methods, 1, 48-92.
Rutstein, A. N., Slater, P. J. B., & Graves, J. A. (2004). Diet quality and resource allocation in the zebra finch. Proceedings of the Royal Society of London. Series B: Biological Sciences, 271(suppl_5), S286-S289.

Validity of the findings

1. [274] This statement is not true as this was not examined systematically in any way.
It’s hard to comment on the validity of the findings before the methodological errors which have been mentioned in the above two sections are addressed.

2. [222-224] Again, this cannot be determined without a valid statistical test and Bonferroni correction.
At several points in your introduction, you justify this research as shedding light on sex allocation [95-97; 110-113] this would be better realized by examining the sex ratio skews at the level of the individual and then controlling for holding, taxon, etc with a more advanced stat test. In order to control for psuedoreplication, you may also want to consider examining at the level of the mother.

3. [338] If this is at the level of species and was not examined systematically, it would be good to have a figure or table that readers could verify this with.

Additional comments

Overall, this is an impressively large dataset and an important topic to address. Your writing is also mostly clear at the level of the sentence and professional but could be more detailed in points. That said, there are large and serious changes which need to be made to your analyses to avoid issues of psuedoreplication. The most important changes would be the use of phylogenetic comparative statistics, followed by using an inferential statistical test to determine skew, consulting up-to-date taxonomical literature when forming the dataset, and finally the use of sex ratio proportions rather than sensu stricto ratios. Finally, your statements could be better supported by the literature and your analysis.

Reviewer 2 ·

Basic reporting

.

Experimental design

.

Validity of the findings

.

Additional comments

Sex allocation is one of the evergreen topics in evolutionary biology and given the current extinction crises it is increasingly interest conservation biologists. Hall et al MS present a well-timed analyses of sex ratios using the ZIMMS (or Species360) database that holds many thousand records of captive species. The dataset is great, the concepts are largely OK (but see below), and the writing is good.

I wish to make the following comments & suggestions.
1. Whilst the various theories of sex allocation look deceptively simple, the devils are in the details. One difficulty is to separate the effects that influence sex ratio bias by individual mothers (eg Trivers Willard, attractive mate, seasonal sex allocation) from the population-level biases (eg sex ratio shifts in offspring sex ratio after wars). Therefore, population-level studies such as Hall et al. cannot possibly test the former set of hypotheses, unless they re-do the analyses at individual parent level.

2. The MS correctly points out the different types of sex ratios (altogether 11 different types, I think), and then seems to focus on the operation sex ratio (OSR). This is problematic since sexual activities vary over both the age of the animal and throughout a year, and thus identifying which individual is “sexually active” seems nearly impossible (see Kokko & Jennions 2008 J Evol Biol).

It looks, most of the analyses used adult sex ratio data (ie ASR), although it is not clear whether the data included young and immature individuals as well (they should not, I think).

3. The domestic species need to be removed from the dataset. I suspect, some poultry farms might be included in the database, and the poultry data can skew the results.

4. The statistical analyses are oversimplistic, and this substantially undermine the conservation value of the study. First, time – i.e., year of a datum - has to me considered in the analyses since captive condition may have changed over time so that influencing the sex ratios. Second, by using ANOVAs, the phylogenetic structure of the data is nearly completely ignored. This inflates the statistical values. A simple way of controlling for phylogeny is mixed-model GLM, although the proper way is phylogenetic comparatiive analyses (such as PGLS). See Cornell & Nakagawa 2017 Current Biol

Phylogenetic control will allow proper comparisons of sex ratios across different IUCN categories using each species’ red list category.

5. The overall male-biased ASR has been well-documented in birds in comparison to other tetrapods. The first study that convincingly reported these was Pipoly et al. 2015 Nature.

Taken together, whilst the motivation of the study is highly applaudable, due to these limitations it cannot really make comparisons between sex ratios in ex-situ and in-situ studies, partly because field-based studies usually report [whichever type of] sex ratios at species level, whereas here potentially 100s of species are blended together under the label of an “order”.

Reviewer 3 ·

Basic reporting

The subject of this paper is relevant to bird conservation worldwide. I was very curious about this topic because I know that the sexual ratios of many species in the world remain uncertain. Thus, I think this work will be useful in the near future.

The manuscript is well-structured and legible, and the objectives of the manuscript are well-defined. However, I wanted to revise the outcomes, but no raw data was shared, only the tables with statistical outcomes.

The authors must follow a scientific page or organization to use the common names. For example, https://birdsoftheworld.org, Clements Checklist of Birds of the World, or Birdlife International Checklist. Also, the authors can actualize their English or Scientific names on these pages.

The literature needs revision, as some references are not in the manuscript (see the revised document) and one citation is not in the references.

Experimental design

It appears that the authors could adopt different approaches by considering their mating systems, whether monogamous (1♂:1♀), polygamous (1♂:≥2♀), or polyandrous(≥2♂:1♀), or some other factors such as parental care, social behaviour, hormone physiology and fitness. However, they decided to use the most straightforward approach (1♂:1♀).

Therefore, to what extent might these analyses be skewed by assuming all species are monogamous with a sex ratio of 1:1? I suggest the authors address this question and include their responses in the methods section to justify why considering a sex ratio of 1:1 is a good approach.

Lines 153-154 The authors mentioned that “Data listings were also removed if there were no known males or females (i.e., all individuals listed as unknown sex) or if there was only a single institution listing” Please explain what they mean by only a single institution listing. In the end, I thought there were just no individuals sexed. Supplementary raw data for all the records may help to re-analyse if necessary.

Line 158 The authors cited that the sex ratio was calculated for each holding as a proportion of males to females. Therefore, they should use for their result sex ratio.

In the methods section, the authors (lines 175-178) do not explain how they calculate or obtain natural sex ratios. Instead, they mentioned that “To determine the effects of housing a species within or outside of their natural range, in-situ distribution as listed by the IUCN (2023) was compared to the country of ex-situ holding (ZIMS) in the Struthioniformes, Columbiformes and Psittaciformes. “. I don´t understand how distribution was used to compare sex ratios within and outside their natural ranges.

Lines 192-194 Explain how the ranges of male-skewed, parity or female-skewed were obtained. Also, here, you need to add citations.

In this section, I was waiting for a table that presented the sex ratio for the taxa. I suggest that the authors present a graph like the one given by Donald (2007).

In lines 282-283, the authors mentioned, “Our study also revealed a male bias pattern in cooperative nesting species that engage nest helpers, which is also supported in the available literature (see Emlen et al. 1986). “But in the result section, no data or analysis is referring to cooperative nesting species. Complement and explain.

Validity of the findings

Their findings depends on their initial condition that was sex ratio 1:1. For that reason, I suggest to justify why use this.

Additional comments

I have made several specific comments on the manuscript. Please, contact the editor if you are not able to download the file.

Annotated reviews are not available for download in order to protect the identity of reviewers who chose to remain anonymous.

---

## Round 0.2 · accepted · Accept

Under normal circumstances, reviews that recommend "major revisions" require a second round of reviews for the corrected manuscript. In this case, I have read the very extensive reviews, felt that they were all technical in nature, and that you have done a good job of answering them. Thank you for such a thorough job,

I see no reason to delay the publication further.